# Asparagine and Glutamine Deprivation Alters Ionizing Radiation Response, Migration and Adhesion of a p53^null^ Colorectal Cancer Cell Line

**DOI:** 10.3390/ijms24032983

**Published:** 2023-02-03

**Authors:** Isabella Guardamagna, Ombretta Iaria, Leonardo Lonati, Alice Mentana, Andrea Previtali, Virginia Uggè, Giovanni Battista Ivaldi, Marco Liotta, Paola Tabarelli de Fatis, Claudia Scotti, Greta Pessino, Maristella Maggi, Giorgio Baiocco

**Affiliations:** 1Laboratory of Radiation Biophysics and Radiobiology, Department of Physics, University of Pavia, 27100 Pavia, Italy; 2Unit of Immunology and General Pathology, Department of Molecular Medicine, University of Pavia, 27100 Pavia, Italy; 3Unit of Radiation Oncology, ICS Maugeri, IRCCS, 27100 Pavia, Italy; 4Unit of Medical Physics, ICS Maugeri, IRCCS, 27100 Pavia, Italy

**Keywords:** colorectal cancer, L-Asparaginase, radiotherapy, ɣH2AX, autophagy, extracellular matrix, adhesion, p53

## Abstract

Colorectal cancer (CRC) is the most prominent form of colon cancer for both incidence (38.7 per 100,000 people) and mortality (13.9 per 100,000 people). CRC’s poor response to standard therapies is linked to its high heterogeneity and complex genetic background. Dysregulation or depletion of the tumor suppressor p53 is involved in CRC transformation and its capability to escape therapy, with p53^null^ cancer subtypes known, in fact, to have a poor prognosis. In such a context, new therapeutic approaches aimed at reducing CRC proliferation must be investigated. In clinical practice, CRC chemotherapy is often combined with radiation therapy with the aim of blocking the expansion of the tumor mass or removing residual cancer cells, though contemporary targeting of amino acid metabolism has not yet been explored. In the present study, we used the p53^null^ Caco-2 model cell line to evaluate the effect of a possible combination of radiation and L-Asparaginase (L-ASNase), a protein drug that blocks cancer proliferation by impairing asparagine and glutamine extracellular supply. When L-ASNase was administered immediately after IR, we observed a reduced proliferative capability, a delay in DNA-damage response and a reduced capability to adhere and migrate. Our data suggest that a correctly timed combination of X-rays and L-ASNase treatment could represent an advantage in CRC therapy.

## 1. Introduction

Colorectal cancer (CRC) is the third most frequent cancer in terms of incidence (with about 1.9 million cases worldwide, corresponding to 10% of all estimated new cancers for both sexes, of all ages, in 2020), but ranks second in mortality (with about ~930,000 deaths, corresponding to 9.5% of all cancer deaths in the same year) (https://gco.iarc.fr/ accessed on 26 October 2022). The continuous increase of CRC cases over recent years is linked to a multifactorial etiology including lifestyle, environmental risk factors and hereditary genetic conditions [1].

The treatment of CRC is often associated with radiotherapy: ionizing radiation (IR, X rays in conventional radiotherapy) is delivered to the target region with the purpose of inducing clonogenic inactivation and death in cancer cells. Radiotherapy (as well as chemotherapy) can be administered to CRC patients either before surgery (neoadjuvant therapy, mainly leading to tumor volume reduction) or following surgery (adjuvant therapy, mainly to remove residual cancer cells and reduce the risk of cancer recurrence). Radiotherapy is currently a fundamental element in the multimodal approach to CRC treatment, particularly for rectal cancers. Over the years, radiotherapy delivery techniques have been greatly improved and different treatment protocols have been established [2]. However, CRC patients’ survival remains poor. Among other factors, this can be attributed to drug resistance [3] and to the high heterogeneity of colorectal cancers. To overcome resistance and improve the outcome of CRC therapy, different treatment strategies need to be explored, including personalized medicine approaches and the use of innovative combinations of treatments that can lead to a clinical advantage, including possible synergistic effects [4].

Generally speaking, advantages in a combined medical treatment occur when the effect of two therapeutic agents (e.g., two different drugs) is greater than the effect seen when each agent is used alone. Though it is hard to have a priori expectations on these kinds of effects, a promising strategy is to look for combinations where the agents have different mechanisms of action and/or different primary targets. As is well known, ionizing radiation acts via DNA and chromosomal damage induction. In this sense, radiotherapy can be successfully combined with the administration of chemotherapy based on antimetabolite drugs, exerting their cytotoxic effects by interfering with DNA synthesis and/or repair. For CRC treatments, Fluoropyrimidine (5-Flourouracil, 5-FU) is the gold standard of first-line treatment: 5-FU acts through the inhibition of cellular thymidylate synthase (TS) and leads to a thymine-less cell death, though 5-FU resistance can be developed and is commonly observed [5]. Among other antimetabolite drugs, L-Asparaginase (L-ASNase) is commonly used in clinical practice as a conventional therapy for acute lymphoblastic leukemia (ALL) [6], and, recently, its possible application in non-hematological tumors has been proposed and investigated [7]. L-ASNase degrades extracellular asparagine and glutamine, two amino acids that are not necessary for healthy cells, but are conditionally essential for cancer cells [8,9,10]. CRC has been indicated as a solid tumor which could potentially benefit from L-ASNase treatment, at least in sensitive subsets [11,12,13]. The combination of radiotherapy and chemotherapy based on L-ASNase is already present in clinical practice, especially in reference to lymphomas [14,15] and might therefore provide a new therapeutic opportunity for CRC.

In this context, we here propose an exploratory in vitro study, using Caco-2 cells (known to be p53^null^) as a model of radioresistant CRC [16], to analyze the combined effects of IR and L-ASNase. In a previous work [16], we have already measured a variety of radiobiological endpoints to characterize the response of Caco-2 cells to X ray doses in the range 0–10 Gy, including dose values of direct therapeutic relevance, e.g., 2 and 5 Gy, typical of normo-fractionated, long-course radiotherapy, and hypo-fractionated, short-course radiotherapy, for CRC in neoadjuvant applications, respectively. Building on previous results, we explored the possible combined effects of X rays and L-ASNase in the same dose range, treating Caco-2 cells with Escherichia coli L-ASNase (EcAII) at a concentration of 1 U/mL. Integrating the results of different endpoints and measurement techniques, we obtained information on the following: clonogenic survival, comparing different time schemes of EcAII administration; distribution in cell cycle phases and regulation of G_2_/M transition; DNA damage induction; regulation of Erk and Akt pathways, involved in mTORC signaling and cell cycle progression [17]; and activation of matrix metalloproteases (MMPs) as an index of invasion potential. Finally, results on the combined effects have been complemented with the measurement of EcAII-induced changes in cell proliferation/migration and adhesion.

## 2. Results

### 2.1. Clonogenic Survival

The survival fraction of Caco-2 cells exposed to different doses of X-rays (0, 1, 3, 5 and 10 Gy, where the 0 sample was sham irradiated) in the absence (w/o) or in the presence of EcAII (w/EcAII) was measured with three different schemes of treatment (Figure 1). In Figure 1A (“pre-IR treatment”) cells were cultured for 72 h w/o or w/EcAII (1 U/mL), then exposed to ionizing radiation (IR) and re-seeded at low density. The colonies that were counted 10 days later were significantly fewer in samples pre-treated with EcAII if compared with the w/o samples at the corresponding IR dose, except for the 1-Gy samples, in which the difference was not significant. In the other two schemes, 1 U/mL EcAII was administered after X-ray exposure, directly at re-seeding (Figure 1B, “post IR treatment”) or 10 days after reseeding (Figure 1C, “treatment after recovery”), with the colonies that were counted at day 13. Caco-2 cells re-seeded in the presence of EcAII immediately post-IR showed a significantly reduced survival fraction at all doses, compared with the untreated group, while, when EcAII was added after 10 d recovery the difference was negligible except for the highest dose of 10 Gy. In the absence of radiation, samples treated only with EcAII showed a reduced clonogenic capability: observed differences are statistically significant with respect to the control (0 Gy, w/o EcAII) in all schemes of treatment.

### 2.2. G_2_/M Phase Transition Regulation

The expression of Chk2 and cyclin B1 (CycB1) levels in Caco-2 cells exposed to different doses of X-rays in the absence (w/o) or in the presence of EcAII (w/EcAII) were measured with the Western blot technique. Based on the results obtained in the clonogenic assay, EcAII was only administered immediately after IR exposure, as it represented the most effective scheme.

Figure 2A,E show representative Western blot images for the densitometric analysis of Chk2 and CycB1, respectively. The signals of the bands of the phosphorylated and total Chk2 protein (p-Chk2 and pan-Chk2, respectively, Figure 2A) were quantified and the phosphorylated fraction calculated. Its changes were evaluated with respect to the untreated sham (set at 100%) of each time point (Figure 2B–D). The CycB1 levels were quantified with respect to glyceraldehyde-3-phosphate dehydrogenase (GAPDH, loading control) (Figure 2E) and the relative changes were calculated with respect to the untreated sham sample (set at 100%) of each time point (Figure 2F–H). Data with normalization and statistical tests for the analysis of radiation effects only (both untreated and EcAII-treated sham set to 100%) are reported in the Appendix A.

It is well known that ionizing radiation induces cell cycle perturbations [18] triggering G_2_/M transition [16] and altering the correct checkpoint-2 signaling pathway. In our dataset this is confirmed by the increase of Chk2 and CycB1 levels that occur after exposure, particularly for CycB1 at the highest 5 Gy dose when compared with the sham condition (Appendix A). Such increases seem to persist at 48 h for Chk2 but not for CycB1. When combining X-ray exposure and EcAII treatment, significant differences were observed only at the highest 5 Gy dose: at 48 h, there is a significant difference in Chk2 activation between w/EcAII and w/o irradiated samples (Figure 2D), while cells irradiated at 5 Gy and treated with EcAII showed a significant difference with w/EcAII irradiated samples as early as 24 h (Figure 2G), as well as a persistent increase in CycB1 at 48 h with respect to the corresponding sham (Appendix A).

A corresponding effect was observed by flow-cytometry for cell cycle analysis (Figure 2I–L). In particular, cells irradiated at 5 Gy and treated with EcAII only (5 Gy w/EcAII, Figure 2L) showed a significant persistent accumulation in the G2/M phase at 48 h if compared with irradiated-only cells (5 Gy w/o, Figure 2L), corresponding to the delay in the rise of CycB1 observed in Western blot in 5 Gy w/EcAII samples. Moreover, a significant reduction in S-phase and accumulation in G1-phase is observed at 24 h in cells treated with EcAII-only (SHAM w/EcAII, Figure 2I).

### 2.3. ɣH2AX Activation

Two different techniques were adopted to analyze DNA damage induced by different doses of X-rays in the absence (w/o) or in the presence of EcAII (w/EcAII).

By means of immunofluorescence, we analyzed the formation of ɣH2AX foci at shortly (30 min and 1 h) after exposure to X-rays to evaluate the initial DNA damage (Figure 3A,B), comparing cells treated with EcAII and the untreated group. To evaluate the DNA damage response in asparagine and glutamine deprivation conditions, cells were pre-treated for 72 h with 1 U/mL EcAII before radiation exposure (0–1–3–5 Gy), to allow for a sufficient time of action of EcAII. From image analysis, we quantified the mean fluorescence intensity (MFI) in arbitrary units (a.u.) per cell associated with the ɣH2AX signal in all investigated conditions (Figure 3C,D). As expected, the signal increases with increasing dose for irradiated samples not pre-treated with EcAII compared with the untreated control (Sham w/o EcAII). When irradiated, samples treated with EcAII seem overall to have an increased ɣH2AX signaling with respect to w/o EcAII ones, but statistical significance is found only at 30 min post-IR for the highest 5 Gy dose. Interestingly, even in the absence of radiation, DNA damage signaling is significantly activated in samples treated for 72 h with EcAII at both time points. In Figure 3E,F the counts of foci/nucleus at 30 min and 1 h after treatment are presented. This analysis does not show significant differences when comparing the samples exposed to X-rays only with those also treated with EcAII. However, a dose-dependent response is observed, with signal saturation between higher doses (Appendix A), for both observed time points, not showing a direct effect of EcAII in foci induction.

Information on residual DNA damage was obtained by means of Western blotting, both without and with EcAII administration. Particularly, to evaluate the long-term DNA damage response in the presence of EcAII, the latter was administered immediately after IR exposure. We evaluated the ratio between the phosphorylated fraction and the total content of the histone variant H2AX, compared with control samples (untreated and non-irradiated), at 6, 24 and 48 h post-IR (Figure 3G). Data with normalization and statistical tests for the analysis of radiation effects only (both untreated and EcAII-treated sham set to 100%) are reported in Appendix A. As expected, if compared with the untreated control, a general increase in ɣH2AX signaling is observed after IR up to 24 h, with a signal normalization or reduction (5 Gy) at 48 h (Appendix A). If normalized for the untreated control (Figure 3H–J), in samples treated with EcAII after radiation exposure, the activation of the signaling seems to have a dose-dependent response, especially at 6 h, with a significant increase at 3 and 5 Gy (Figure 3H). At 24 h, only the difference between the control and samples treated with EcAII still remains significant and a non-significant trend to signal increase is observed up to 48 h (Appendix A and Figure 3H–J). Taken all together, therefore, immunofluorescence and Western blot data from unirradiated samples indicate an EcAII-induced DNA damage response starting a short time after administration and lasting up to 24 h, while the initial increase in ɣH2AX signaling when samples are both EcAII-treated and irradiated with the highest 5 Gy dose seems to definitively vanish starting from 6 h post IR.

### 2.4. Erk and Akt Regulation

The phosphorylation status of Erk and Akt was analyzed via Western blot to evaluate the metabolic response of cells exposed to different doses of X-rays without (w/o) or with EcAII treatment (w/EcAII). EcAII was administered after radiation, and samples were analyzed at 6, 24 and 48 h after IR exposure (Figure 4A,E). Data are presented as the ratio between the intensity of the band of the phosphorylated protein and that of the total protein, evaluating the activation change with respect to the untreated sham (set at 100%) at each time point. Data with normalization and statistical tests for the analysis of radiation effects only (both untreated and EcAII-treated sham set to 100%) are reported in Appendix A. Treatment with radiation has a significant effect on the state of Erk phosphorylation in the short-term only for the highest 5 Gy dose, but a positive trend can be observed. A significant increase in Erk phosphorylation levels was instead evident at 48 h after radiation and was maintained in the presence of EcAII (Appendix A). In Figure 4B–D, samples not exposed to radiation but treated with EcAII, show a significant increase in Erk at all the investigated time points. Samples exposed to radiation and treated with EcAII generally present higher Erk levels with respect to untreated ones, but statistically significant differences are observed only at 6 h at a dose of 5 Gy dose, and at 24 h at dose of 1 Gy. Akt levels (Figure 4F–H), as expected, seem to be reduced in samples treated with EcAII. This is true for non-irradiated samples at 24 h and 48 h, while the response observed at 6 h at the highest 5 Gy dose seems to persist up to 24 h and, at the same time point, samples exposed to 3 Gy showed a significant reduction in the signal (Appendix A).

### 2.5. Extracellular Metalloproteinases Regulation, Migration and Adhesion

Gelatin zymography was used to evaluate the gelatinolytic activity of MMP-9 in response to different doses of X rays without (w/o) and with EcAII (w/EcAII) treatment (Figure 5A). EcAII was administered after irradiation, and culture media were analyzed at 6 and 24 h after IR exposure. The densitometric quantification (Figure 5B,C) was performed by expressing the intensity of white bands with respect to the unirradiated and untreated control sample for each time point. In samples treated with EcAII and exposed to ionizing radiation, there is a significant reduction of MMP-9 gelatinolytic activity at 5 Gy at both time points and at 3 Gy at 6 h. In the samples treated with EcAII, but not exposed to radiation, however, an increase in the activity of MMP-9 was observed for both time points.

To better interpret the MMP-9 activity increase in unirradiated samples treated with EcAII, we performed a wound healing assay, evaluating the time required for the closure of the gap after removal of the insert, both for untreated cells and for cells with EcAII treatment (with EcAII administration at the time of insert removal, 24 h after seeding). Representative images are shown in Figure 6A. Results reported in Figure 6B are expressed as percentage of the gap area not covered by cells as a function of time (days) after insert removal. In untreated samples, the area is totally covered between day 4 and 5. Cells treated with EcAII show a significant slowdown in their ability to proliferate and migrate to cover the area left free by the insert: differences are observed as early as day 1, and become statistically significant from day 3 onwards, with full coverage not reached within the same time frame of 5 d.

A similar delay in cell adhesion capability is observed in RTCA (Figure 6C). Indeed, Caco-2 cells treated with 1 U/mL EcAII have a delta CI significantly lower than control samples at 48 h after treatment (Figure 6D), indicating a reduction of cell proliferation and adhesion compared with the untreated sample.

## 3. Discussion

The search for new therapeutic approaches, both to successfully eradicate primary cancers and to prevent relapses and metastases, is of utmost importance for oncology [19]. IRs have long been used in clinical practice for the treatment of various cancers; in fact, radiation therapy is considered crucial for the primary management of locally advanced CRC and can essentially contribute to treatments in locally recurrent, oligometastatic patients or to palliative care [20]. L-ASNase has, instead, long been used as a treatment for ALL [21] and, only recently, has been proposed as a treatment for solid tumors, with its effects and underlying mechanisms of action being now a novel field of investigation [22]. L-ASNase acts at the tumor micro-environment level, by removing asparagine and glutamine extracellular supply. Cancer cells require a nutrient-rich environment to proliferate and migrate. The shortage of asparagine and glutamine mediated by L-ASNase causes a slowdown in essential cellular processes such as protein synthesis, DNA replication, DNA repair and, in a more general view, proliferation by repression of the Akt-mTOR signaling pathway. Moreover, removal of amino acids impairs cells’ capability to produce glutathione, causing an increase in ROS-mediated mitochondria and DNA damage [17].

CRC is among the solid tumors with the potential to benefit from L-ASNase treatment, as metabolic reprogramming with amino acid dependence has been described to be a major feature of this tumor type [11,23]. In this context, the purpose of this work was to investigate the effects induced by the combination of X-ray exposure and nutrient deprivation obtained with EcAII, using Caco-2 cells as an in vitro model for colorectal adenocarcinoma cells. In vitro models offer an extremely useful first-step tool for the exploration of possible synergistic effects between different therapeutic agents in controlled experimental conditions and for varying, e.g., the radiation dose, drug dose, or administration scheme (fractionation, time intervals and time order). Caco-2 cells were chosen for this study, as they have already been well-characterized in terms of their radiation response to X ray doses in the range 0–10 Gy [16]. Additionally, their p53^null^ status makes them an interesting model to study response mechanisms to therapeutic agents, as p53^null^ CRC subtypes are often resistant to canonical therapies and have a poor prognosis [24].

Clonogenic survival was considered the starting point for this study, as inducing clonogenic inactivation in cancer cells is a key goal of any oncological treatment. Our new data on Caco-2 cell survival after IR only are consistent with what has been previously observed [16] and a combined effect of X rays and EcAII can, in fact, be appreciated from the data reported in Figure 1. EcAII administration immediately before or after radiation exposure results in a significant reduction in survival at all doses compared with EcAII alone and IR alone. The only exception is the 1 Gy condition in the pre-IR treatment scheme. This observation is intriguing, and a possible explanation could reside in an adaptive response mechanism [25]: cells pre-treated with EcAII, with activated DNA damage signaling and repair pathways at the time of the irradiation, could prove more resistant to the medium-low challenge dose of 1 Gy. This hypothesis should be tested with dedicated measurements at X-ray doses lower than those usually applied for radiation therapy treatments, and such measurements therefore fall outside the scope of this work. Most importantly, when the EcAII treatment is carried out 10 days after low-density re-seeding, no further reduction in clonogenic survival is observed with respect to what can be ascribed to radiation alone. It is reasonable to believe that the selection of the most resistant cells, those able to recover from the radiation insult and to create colonies, might have a role because if clones with a better adaptive capacity are selected, these prove to not be further affected by EcAII administration. In the perspective of therapeutic applications of a combined radiation–L-ASNase treatment, this finding suggests careful consideration of the timing for the recovery of target tissues after the X-ray dose. Schemes with EcAII administered prior to or immediately after radiation, in fact, seem to be critical to favoring the efficacy of the treatment combination. In contrast, and based on our results, treatment of residual cancer cells with L-ASNase after first-line radiotherapy application, might not lead to a clinical advantage. Therefore, only treatment schemes with L-ASNase administered immediately before or after radiation exposures were considered for the other endpoints investigated in this work.

To gain insight into mechanisms affected by the combination of treatments, a selection of molecular pathways that are typically regulated by IR and/or EcAII were investigated; in particular, the proteins Chk2 and CycB1, involved in the regulation of the transition of the G_2_/M phases, were analyzed. Modifications in Chk2 and CycB1 signaling after radiation alone confirmed previous findings [16]. In the presence of nutrient deprivation induced by EcAII, the effects of the combination with X-rays on these two proteins are more difficult to observe, and seem to be limited to the highest 5 Gy dose only. For irradiated and EcAII-treated cells, results—such as the higher Chk2 signal at 48 h, together with the lower CycB1 signal at 24 h and persistent CycB1 increase at 48 h—with respect to the unirradiated control, can in general be interpreted as being due to a slowdown in the activation of signaling to induce the G_2_ phase arrest, associated with a longer arrest persistence. This observation was confirmed also by cell cycle phase analysis with flow-cytometry in CaCo-2 cells irradiated at 5 Gy. Indeed, 48 h after radiation, G2 arrest is still evident in EcAII-treated cells and is significantly higher than in cells treated with radiation only.

Both perturbation in cell-cycle progression and clonogenic inactivation are known to be possibly correlated with DNA damage induction, delays/arrests due to the activation of DNA repair mechanisms, and finally misrepair or repair failure leading to cell death. Activation of DNA damage and repair signaling was therefore measured in the investigated conditions. The EcAII treatment alone induces DNA damage signaling via the phosphorylation of H2AX histones that, combining with results from pan-H2AX immunofluorescence and Western blotting, seems to persist up to at least 24 h. It is known that EcAII treatment induces ROS production, and that this can in turn lead to indirect DNA damage [26]. In samples irradiated and treated with EcAII, an increase in the ɣH2AX signal can be observed, although only significant at the highest 5 Gy dose, and the effect of the combination seems to be limited to the first hours after the exposure, with no differences observed after 6 h onwards, most likely because of the adaptation of tumor cells to the environmental changes. Moreover, analysis of γH2AX foci does not show any difference in EcAII-treated samples vs. untreated ones and this is expected when the EcAII treatment is applied, given the prevalence of damage in the form of ssDNA breaks due to ROS production. Indeed, pan-H2AX measurement allows the analysis of total DNA damage, while analysis of foci-like structures is, instead, focused on dsDNA breaks because of the formations of foci-like structures given by the signal amplification at dsDNA sites [27,28,29,30]. Altogether, and in the absence of evidence of significant lethal replicative stress and apoptotic cells, the difference we measure in terms of mean γH2AX fluorescence intensity points at the predominance of EcAII-induced ssDNA breaks with respect to untreated samples, though this aspect can be better described by dedicated quantitative assays such as single-cell gel electrophoresis.

Another important effect of asparagine and glutamine deprivation is the activation of pro-autophagic pathways mediated by the phosphorylation of Erk that acts downstream on LC3 II expression levels [31]. In our experimental setting, treatment of Caco-2 cells with L-ASNase significantly increased Erk phosphorylation levels at all the analyzed time points, suggesting there is also a prompt pro-autophagic effect of EcAII in this cell line. Additionally, treatment with radiation only leads to higher Erk phosphorylation levels, particularly at 48 h after irradiation. Very likely, in this case, activation of Erk pathways mediates the cell response to DNA damage, activating DNA repair. The generally higher levels of phosphorylation of Erk when cells are irradiated and treated with L-ASNase translate into statistically significant differences only for a few dose and time points, with no clear pattern. A possible explanation might be that the increase in Erk phosphorylation mediated by DNA damage makes L-ASNase-induced changes less evident with respect to cells treated with L-ASNase only.

Repression of PI3/Akt signaling is a hallmark of L-ASNase-mediated metabolic distress [32]. In cells exposed to L-ASNase, the reduction in pAkt results in the downstream blockade of mTORC1, leading to inhibition of proliferation and activation of pro-autophagic and pro-apoptotic signaling. In Caco-2 cells, as expected in CRC cells, the PI3/Akt pathway is constitutively activated and promotes cell proliferation, adhesion and migration. As previously described in radioresistant colon cancer cell lines [33], response to radiation stimulates the activation of Akt. In our experimental model, no significant increase in Akt phosphorylation can be observed after radiation, but a trend in this direction can be observed after X-ray exposure at all doses at 24 h. Interestingly, co-treatment with EcAII seems to invert this trend, with a significant repression of the pathway activation that can be observed mainly at 3 and 5 Gy, 24 h post-IR. Moreover, treatment with EcAII causes a significant reduction in pAKT, and this modification at the molecular level could lead to macroscopic changes in the cell population behavior. This has been confirmed in terms of a reduction in proliferation/migration and adhesion by independent measurements with the wound healing assay and the iCELLigence RTCA system.

Activation of the PI3/Akt signaling pathway also plays an important role in the regulation of matrix metalloproteinases (MMPs); in particular, MMP-9 has been seen to be regulated by Akt [34,35]. Data presented in this work suggest a correlation between the significant reduction of the Akt signal and the reduction of MMP-9 when EcAII is administered to irradiated samples. Surprisingly, the unirradiated sample treated with EcAII showed no significant alteration of Akt at 6 h, and a significant decrease in Akt at 24 h, which, however, does not correspond to a decrease in MMP-9. From zymography data, instead, an increase of MMP-9 is observed in unirradiated samples treated with EcAII. This could be explained by the fact that the deprivation of Asp and Gln also causes production of ROS with consequent induction of DNA damage, which activates MMPs from the zymogen state [36]. The remodeling activity induced by the MMP-9 in the unirradiated control suggested investigating behavioral properties of the cell population with independent measurements by means of techniques such as the wound-healing assay and the iCELLigence RTCA system device. The increase in MMP-9 supports the observation of the reduced adhesion measured with the iCelligence. However, the reduced adhesion capability does not correspond to an increase in migration, as demonstrated by the wound healing assay, thus also suggesting a possible induction of death mechanisms such as anoikis.

## 4. Materials and Methods

### 4.1. Cell Culture and Irradiation Protocols

Caco-2 cells (American Tissue Culture Collection, ATCC, Manassas, VA, USA) were cultured in Dulbecco’s Modified Eagle’s medium (DMEM (EuroClone, Siziano, Italy)) supplemented with 10% fetal bovine serum (FBS (EuroClone, Siziano, Italy)), 2 mM L-glutamine (EuroClone, Siziano, Italy), 100 U/mL penicillin, and 100 mg/mL streptomycin (EuroClone, Siziano, Italy) at 37 °C in a humidified atmosphere with 5% CO_2_. Caco-2 cells were at passage 15th to 35th for all experiments. Irradiations were performed at the radiotherapy department of Istituto di Ricovero e Cura a Carattere Scientifico (IRCCS) S. Maugeri (Pavia, Italy) with a linear accelerator routinely used for radiotherapy treatment. The reference radiation doses to which the cells have been exposed were: 0 (Sham), 1, 3, 5, 10 Gy, or other doses specifically indicated for each of the experiments. Irradiations were performed as previously described in detail [37].

### 4.2. Survival Assay

The clonogenic survival of Caco-2 cells was evaluated by plating cells in three different combinations of treatments between X-ray exposure (0, 1, 3, 5, 10 Gy) and EcAII (1 U/mL). In the “pre-IR” treatment scheme, 24 h after seeding 1.5 × 10^6^ cells in a T75 culture flask, 1 U/mL EcAII was administered for 72 h before X-ray exposure. After radiation exposure, cells were re-seeded at low density in a T25 flask in the presence of 1 U/mL EcAII in the L-ASNase pre-treated samples (w/EcAII). In the “post-IR” treatment scheme, 24 h after seeding at 1.5 × 10^6^ in T75 culture flask, cells were irradiated and re-seeded at low density in T25 flask. In w/EcAII samples 1 U/mL EcAII was added at re-seeding. In a third treatment scheme, colonies grew after 10 d w/o EcAII and were cultured for another 72 h in the presence of 1 U/mL EcAII. For all the experiments, a group exposed to X-ray radiations but without EcAII administration was considered as an “untreated control”. Ten or 13 d after reseeding, colonies were fixed and stained with a solution containing 1% Crystal Violet (Sigma-Aldrich, St. Louis, MO, USA). The day after, colonies were counted by a colony counter (SC6Plus, Stuart, Cernusco sul Naviglio, Italy).

### 4.3. Cell Cycle Analysis

For cell cycle analysis, Caco-2 cells were seeded at 2.5 × 10^5^ cells in T25 Flasks (Greiner-Bio One, Kremsmünster, Austria) 72 h before radiation exposure at a selected dose (5 Gy). Immediately after X-ray exposure, the medium was replaced, including 1 U/mL EcAII only in the EcAII treated group. Before each timepoint (24 and 48 h), cells were incubated with 2 μg/μL 5-ethynyl-2′-deoxyuridine (EdU) [34] for 1 h, then fixed following the manufacturer’s instructions with minor modifications. Briefly, cells were harvested by trypsinization and fixed in 4% w/v paraformaldehyde (PF) for 5 min, then permeabilized in 70% EtOH in 0.9% NaCl in ddH2O. EdU detection was revealed by Click-iT Plus EdU Alexa Fluor488 Flow Cytometry Assay Kit (Invitrogen, Waltham, MA, USA) and the DNA content was measured by FxCycle Violet dye (4′,6-diamidino-2-phenylindole, dihydrochloride, Invitrogen).

The Attune NxT Acoustic Focusing flow cytometer (ThermoFisher Scientific, Waltham, MA, USA) employed for these experiments. All analyses were performed with Attune NxT software v 4.2.1627.1. Data are represented as a percentage of each cell cycle phase.

### 4.4. Western Blot

An amount of 2.5 × 10^5^ Caco-2 cells were cultured in T25 flasks for 72 h, then irradiated and treated with 1 U/mL EcAII (w/EcAII samples) or in normal medium (w/o samples) after radiation exposure. Cells were collected by trypsinization at 6, 24 and 48 h after treatments, centrifuged at 300 g for 3 min at room temperature (RT), washed in phosphate buffer saline, and centrifuged again at 3400 g for 5 min at RT. The resulting pellets were stored at −80 °C.

To detect H2AX, Chk2 and CycB1 proteins, pellets were resuspended in 100 μL lysis buffer (10 mM Tris-HCl, 2.5 mM MgCl_2_, 0.5% Triton X-100, 1 mM PMSF (nuclear extraction kit, Cayman Chemical, Ann Arbor, MI, USA), 2.5 U/μL Benzonase^®^ (Merck KGaA, Darmstadt, Germany) and incubated by shaking on ice for 30 min. After lysis, samples were quantified with the Bradford (VWR) method. For each sample, 30 μg of proteins were mixed with a 3× SDS-loading buffer (65 mM Tris-HCl pH 7.4, 100 mM DTT, 10% glycerol, 1% SDS, 0.02% Bromophenol blue) and loaded onto a 10% or 12% acrylamide/bis-acrylamide gel for SDS-PAGE electrophoretic separation. Proteins were electrotransferred to nitrocellulose 0.2 μm membranes and blotted with the intended primary antibodies: anti-Chk2 (dilution 1:1000, Cell Signaling, Danvers, MA, USA (RRID:AB_2229490)), anti-P-Chk2 (T68) (dilution 1:1000, Cell Signaling (RRID:AB_331479)), anti-H2A.X (dilution 1:1000, Cell Signaling (RRID:AB_10860771)), anti-P-H2A.X (S139) (dilution 1:1000, Abcam (RRID:AB_1640564)), anti-CyclinB1 (dilution 1:1000, Cell Signaling (RRID:AB_2233956)) and anti-GAPDH (dilution 1:1000, Cell Signaling (RRID:AB_10622025)). The secondary HRP-conjugated antibodies that were used were sheep anti-mouse IgG (dilution 1:2000, GE Healthcare, Chicago, IL, USA (RRID:AB_772210)) and donkey anti-rabbit IgG (dilution 1:2000, GE Healthcare (RRID:AB_772206)). To detect Akt and Erk, cell pellets were incubated in 100 μL RIPA buffer (30 mM HEPES, pH 7.4, 150 mM NaCl, 1% Nonidet P-40, 0.5% sodium deoxycholate, 0.1% sodium dodecyl sulfate, 5 mM EDTA) added with protease (Complete™, Mini, EDTA-free, Roche, Basilea, Switzerland) and phosphatase (Phosphatase Inhibitor Cocktail 2, Sigma-Aldrich, St. Louis, MO, USA) inhibitors. After a 20 min incubation on ice in mild shaking, samples were centrifuged (10,000 rpm, 4 °C, 15 min,) to remove cell debris. Then, the total protein content was measured by bicinchoninic acid assay and 30 µg proteins per well were loaded onto a 12% acrylamide/bis-acrylamide gel for SDS-PAGE electrophoretic separation. Proteins were transferred onto a PVDF 0.22 μm membrane and incubated with the required primary antibodies: anti-Akt (dilution 1:1000, Cell Signaling, Danvers, MA, USA (RRID:AB_915783)); anti-p-Akt (dilution 1:1000, Cell Signaling, Danvers, MA, USA (RRID:AB_2315049)); anti-Erk (dilution 1:1000, Cell Signaling, Danvers, MA, USA (RRID:AB_390779)); and anti-p-Erk (dilution 1:1000, Cell Signaling, Danvers, MA, USA (RRID:AB_11127856)).

### 4.5. Immunofluorescence

To analyze the DNA damage response, Caco-2 cells (1.5 × 10^5^) were seeded on coverslips and the following day treated with 1 U/mL EcAII for 24 h, then exposed to X-rays. At 30 min and 1 h after exposure, samples were fixed in 4% formaldehyde and permeabilized in 70% ethanol diluted in ddH_2_O. Coverslips were incubated with blocking solution (1% BSA in PBT) for 30 min at RT, then with the primary antibody anti-phospho H2AX (S139) (dilution 1:400, Cell Signaling Technologies (RRID:AB_2118010)) for 1 h at RT and the secondary antibody goat anti-rabbit IgG 555 (dilution 1:200, Molecular Probes (RRID:AB_2535851)) for 30 min at RT. Finally, they were washed with Hoechst 33342 dye (Abnova, Taipei, Taiwan) and mounted with Mowiol (Calbiochem, San Diego, CA, USA) containing 0.25% 1,4-diazabicyclo-octane (Sigma-Aldrich, St. Louis, MO, USA) as antifading agent. γH2AX signal was visualized with fluorescent microscopy (Olympus BX51, Olympus, Shinjuku, Tokyo, Japan). Images were acquired by digital CCD camera (Retiga-2000R) and the mean fluorescence intensity (MFI) was quantified by ImageJ v 1.53 [38] calculating the corrected total cell fluorescence (CTCF) expressed as:CTCF = Integrated Density − (Area of selected cell X Mean fluorescence of background readings).

γH2AX foci were scored by ImageJ v 1.53 with the analyze particles function, setting the range 0–150 pixel^2^ as limit to define a single focus with a diameter of 0.5 μm.

### 4.6. Gelatin Zymography

Measurements of matrix metalloproteinases (MMP-9) in the culture medium were performed following the experimental procedure already published in [39], with minor changes. Conditioned media (500 μL, from samples used for Western blotting analysis) were collected, centrifuged at 4600 g (Thermo Scientific CL31R, Waltham, MA, USA) and supernatants mixed in sample buffer 2× (0.5 M Tris-HCl pH 6.8, 20% glycerol, 10% SDS, 0.1% Bromophenol blue), ratio 1:1, and stored at −80 °C. Amounts of 20 μL of each sample were loaded on a 10% polyacrylamide gel containing 1 mg/mL bovine type B gelatin (Sigma-Aldrich, St. Louis, MO, USA). Gels were stained with Coomassie Blue R-250 (0.5% *w/v*) and subsequently de-stained and acquired with Image Gel Analyzer (Bio-Rad, Hercules, CA, USA).

### 4.7. Wound Healing

An amount of 7 × 10^4^ cells were seeded in each insert (100 µL) of the Ibidi Culture-Insert (Madison, WI, USA) and cultured at 37 °C. After 24 h, the insert was removed, creating a cell-free gap and 1 U/mL EcAII were added in the treated samples. Wound closure was monitored daily and pictures acquired from 0 to 5 days with a Leica DMIRB inverted light microscope equipped with a Leica DFC310 FX camera (Leica, Wetzlar, Germany). The area quantification of the gap was performed using the software ImageJ v 1.53 [38].

### 4.8. Real-Time Cell Adhesion Analysis

Real-time cell analysis (RTCA) was performed using the ACEA iCELLigence device (ACEA Biosciences, San Diego, CA, USA) [40].

Cells were seeded at 10^5^ per well in 650 μL complete medium in two 8-well iCELLigence dedicated plates. After 30 min incubation at room temperature, the plates were incubated in the iCELLigence instrument at 37 °C, 5% CO_2_ in a humidified environment. Impedance was read every minute for 72 h. After 24 h, 1 U/mL EcAII dissolved in medium was added to the treated samples and an equal volume of medium without EcAII was added to the untreated samples.

Data analysis was undertaken by using the cell index (CI), a non-dimensional parameter directly dependent on the impedance variation, calculated between the beginning of the treatment (24 h after seeding) and the end of the experiment (72 h after seeding). The deltaCI was expressed as a percentage of CI72h vs. CI24h. Data are an average of eight replicates obtained from at least two independent experiments.

### 4.9. Recombinant Asparaginase

Recombinant N24S *E. coli* type II Asparaginase (EcAII) was used for all the experiments requiring L-ASNase treatment. Recombinant EcAII was produced and purified as reported in Maggi et al., 2017 [8]. After purification, the protein was lyophilized and, when needed, resuspended in culture media in sterile conditions at the needed concentration for the treatment.

### 4.10. Statistical Analysis

For the different endpoints, each experimental value represents the mean of at least three independent measurements (biological replicates); errors are given as standard deviation (SD) (details are specified in figure captions). The statistical significance (*p*-value) was calculated by means of the two-tailed multiple Student’s *t*-test. Details are given in the captions of the relevant figures.

## 5. Conclusions

The present study shows a positive correlation between the in vitro cytostatic activity of X-ray and L-ASNase, two well established therapeutic approaches that have never been combined in in vitro or in vivo models for the treatment of colon cancer. In our in vitro experimental model, Caco-2 cells, a p53^null^ CRC cell line with poor response to classical chemotherapy and radiations, show a reduced proliferative index if exposed to X-rays in an amino acid deprivation state mediated by L-ASNase. In particular, the observed reduced clonogenic activity in the presence of L-ASNase can be linked to a less efficient DNA damage response, to the activation of the Erk-mediated autophagic process, to the reduction of Akt-mediated proliferation signaling, and to a reduced capability of the cells to adhere and migrate.

Further specific experiments will be required, and mathematical models shall be applied, to establish the details of the quantitative interactions between radiation and L-ASNase as therapeutic agents, to verify if the desired effect of synergy beyond additivity is attained. Taken together, these observations, however, pave the way for further studies on the combined effect of amino acid deprivation and X-rays in other CRC cell lines of different genetic background, and, prospectively, in the treatment of highly aggressive and poorly treatable CRC, such as p53^null^ subtypes.

## Figures and Tables

**Figure 1 ijms-24-02983-f001:**
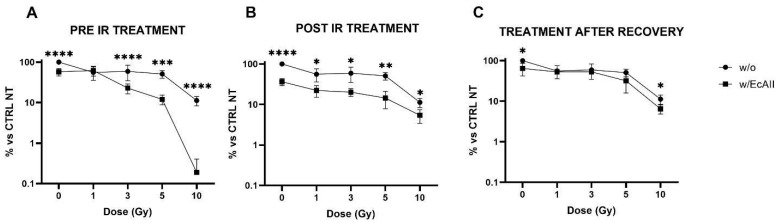
Clonogenic survival assay for Caco-2 cells exposed to different doses of X rays (0, 1, 3, 5 and 10 Gy), without (w/o, black bars) or with 1 U/mL EcAII (w/EcAII, gray bars) treatment, with different treatment schemes. Data are expressed as the percentage of colonies counted with respect to the unirradiated and untreated sample (% vs. CTRL NT). Data are the mean ± SD obtained from at least three independent experiments. Panel (**A**), Caco-2 cells treated with EcAII before X-ray exposure, colonies counted at day 10. Panel (**B**), Caco-2 cells treated with EcAII immediately after X-ray exposure, colonies counted at day 10. Panel (**C**), Caco-2 cells treated with EcAII after 10 days from X-ray exposure, colonies counted at day 13. Statistical significance (Student’s *t*-test) was calculated comparing the w/o and w/EcAII conditions for each dose, and reported as follows: * *p* < 0.05, ** *p* < 0.01, *** *p* < 0.001, **** *p* < 0.0001.

**Figure 2 ijms-24-02983-f002:**
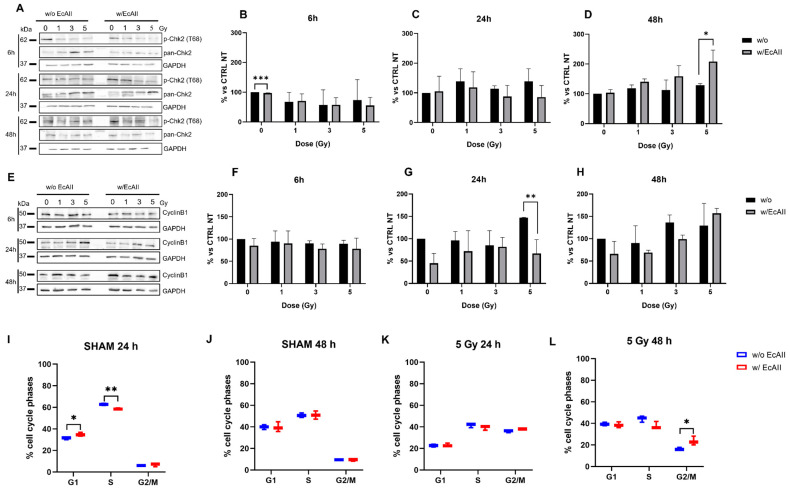
G_2_/M transition phase protein analysis for Caco-2 cells exposed to different doses of X rays (0, 1, 3 and 5 Gy), without (w/o) or with 1 U/mL EcAII (w/EcAII) administered after radiation and cell cycle analysis of CaCo-2 cells exposed to different doses of X rays (0, 5 Gy), without (w/o) or with 1 U/mL EcAII (w/EcAII) administered after radiation. Panel (**A**), Illustrative Western blot images of phosphorylated Chk2 (p-Chk2 (T68)) and pan-Chk2. The densitometric analysis was performed by evaluating the p-Chk2 (T68)/pan-Chk2 ratio, with changes expressed relative to the unirradiated and w/o control at each time point (% vs. CTRL NT). Samples in the absence of EcAII (black bar) and with EcAII (gray bar) were harvested at: Panel (**B**), 6 h after irradiation; Panel (**C**), 24 h after irradiation; Panel (**D**), 48 h after irradiation; Panel (**E**), illustrative Western blot analysis images of Cyclin B1. The densitometric analysis was performed by evaluating the CycB1/GAPDH (loading control) ratio, with changes expressed relative to the unirradiated and w/o control at each time point (% vs. CTRL NT). Samples in the absence of EcAII (black bar) and with EcAII (gray bar) were harvested at: (**F**), 6 h after irradiation; (**G**), 24 h after irradiation; (**H**), 48 h after irradiation. Panel (**I**), percentage of cell cycle phases at 0 Gy w/o or with EcAII at 24 h. Panel (**J**), percentage of cell cycle phases at 0 Gy w/o or with EcAII at 48 h. Panel (**K**), percentage of cell cycle phases at 5 Gy w/o or with EcAII at 24 h. Panel (**L**), percentage of cell cycle phases at 5 Gy w/o or with EcAII at 48 h. Data reported are mean ± SD, obtained from at least three independent experiments. Statistical significance (Student’s *t*-test) was calculated comparing the w/o and w/EcAII conditions for each dose and time point, and reported as follows: * *p* < 0.05, ** *p* < 0.01, *** *p* < 0.001.

**Figure 3 ijms-24-02983-f003:**
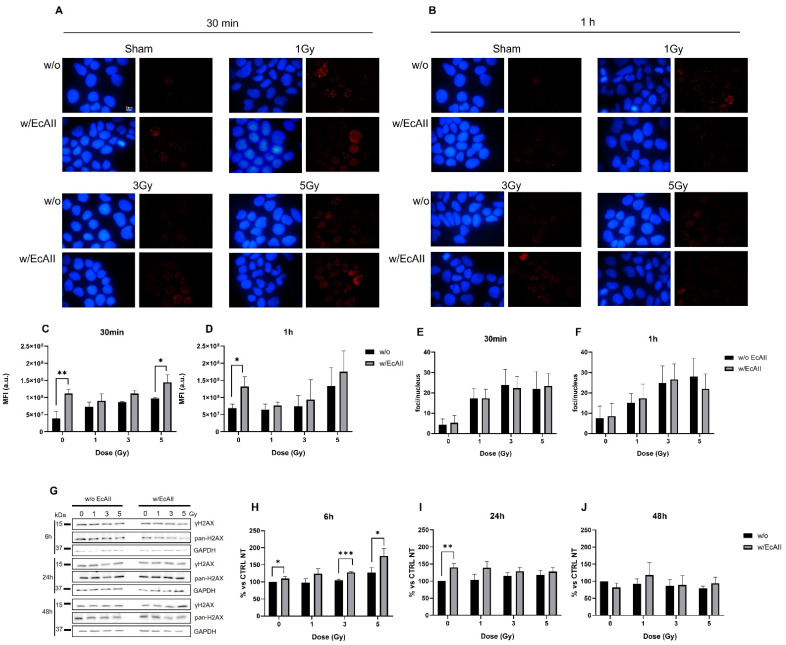
ɣH2AX analysis for Caco-2 cells exposed to different doses of X rays (0, 1, 3 and 5 Gy), without (w/o) or with 1 U/mL EcAII (w/EcAII). Panels (**A**,**B**), illustrative fluorescence microscopy images for the different irradiation and treatment conditions obtained with Hoechst (blue for nuclear DNA, left) and ɣH2AX (red for DNA damage foci, right) of the sample fixed 30 min and 1 h after irradiation, respectively. Images were acquired using a 100× magnification. Cells pre-treated with EcAII for 72 h before radiation exposure. Panels (**C**,**D**), mean fluorescence intensity (MFI) per cell of the ɣH2AX signal (a.u.). (**E**,**F**) The count of H2AX foci/nucleus was obtained using open software ImageJ v 1.53 for the sample harvested 30 min and 1 h after irradiation, respectively. (**G**) Cells treated with EcAII after radiation exposure. Illustrative Western blot images of ɣH2AX (S139) and pan-H2AX. The densitometric analysis was performed by evaluating the p-H2AX/total H2AX ratio with changes expressed relative to the unirradiated and untreated control at each time point (% vs. CTRL NT). Samples in the absence of EcAII and with EcAII were harvested at 6 h (**H**), 24 h (**I**) and 48 h after irradiation (**J**). Data reported are mean ± SD, obtained from at least three independent experiments. Statistical significance (Student’s *t*-test) was calculated comparing the w/o and w/EcAII conditions for each dose and time point (Student’s *t*-test) and is as follows: * *p* < 0.05, ** *p* < 0.01, *** *p* < 0.001. In all charts black bars represent samples w/o EcAII and gray bars represent samples w/EcAII.

**Figure 4 ijms-24-02983-f004:**
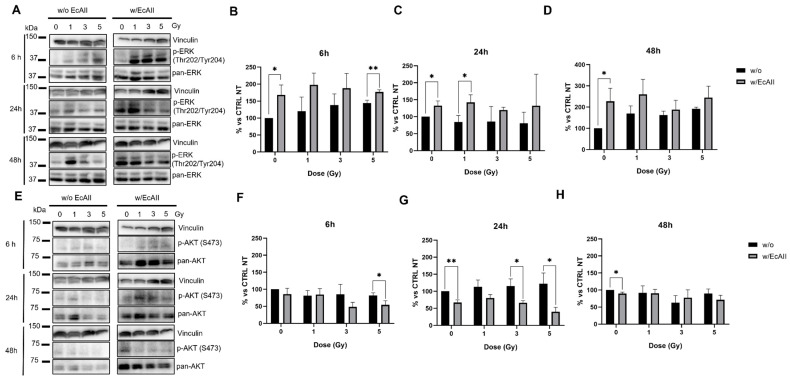
Metabolic response evaluation for Caco-2 cells exposed to different doses of X rays (0, 1, 3 and 5 Gy), without (w/o) or with 1 U/mL EcAII (w/EcAII) administered after radiation. (**A**) Illustrative Western blot images of phosphorylated ERK (T202/Y204) and pan-ERK. The densitometric analysis was performed by evaluating the p-ERK/total ERK ratio, with changes expressed relative to the unirradiated and untreated control at each time point (% vs. CTRL NT). Samples w/o EcAII (black bar) and with EcAII (gray bar) were harvested: (**B**) 6 h after irradiation; (**C**) 24 h after irradiation; and (**D**) 48 h after irradiation. (**E**) Illustrative Western blot images of phosphorylated AKT (S473) and pan-AKT. The densitometric analysis was performed by evaluating the p-AKT/total AKT ratio, with changes expressed relative to the unirradiated and untreated control at each time point (% vs. CTRL NT). Samples in the absence of EcAII (black bar) and with EcAII (gray bar) were harvested: (**F**) 6 h after irradiation; (**G**) 24 h after irradiation; (**H**) and 48 h after irradiation. Data reported are mean ± SD, obtained from at least three independent experiments. Statistical significance (Student’s *t*-test) was calculated by comparing the w/o and w/EcAII conditions for each dose and time point and is reported as follows: * *p* < 0.05, ** *p* < 0.01.

**Figure 5 ijms-24-02983-f005:**
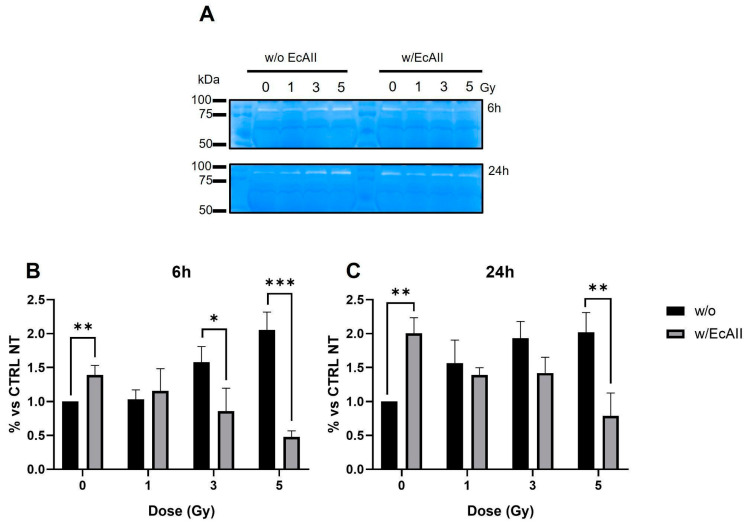
MMP-9 activity with gelatin zymography for Caco-2 cells exposed to different doses of X rays (0, 1, 3 and 5 Gy), without (w/o) or with 1 U/mL EcAII (w/EcAII). (**A**) Representative images of gelatin zymography with identification of MMP-9 bands. Changes in MMP-9 gelatinolytic activity were expressed relative to the unirradiated and untreated control at each time point (% vs. CTRL NT) and quantified for samples without (w/o) of EcAII (black bar) and with EcAII (gray bar): (**B**) 6 h irradiation; (**C**) 24 h after irradiation. * *p* < 0.05, ** *p* < 0.01, *** *p* < 0.001.

**Figure 6 ijms-24-02983-f006:**
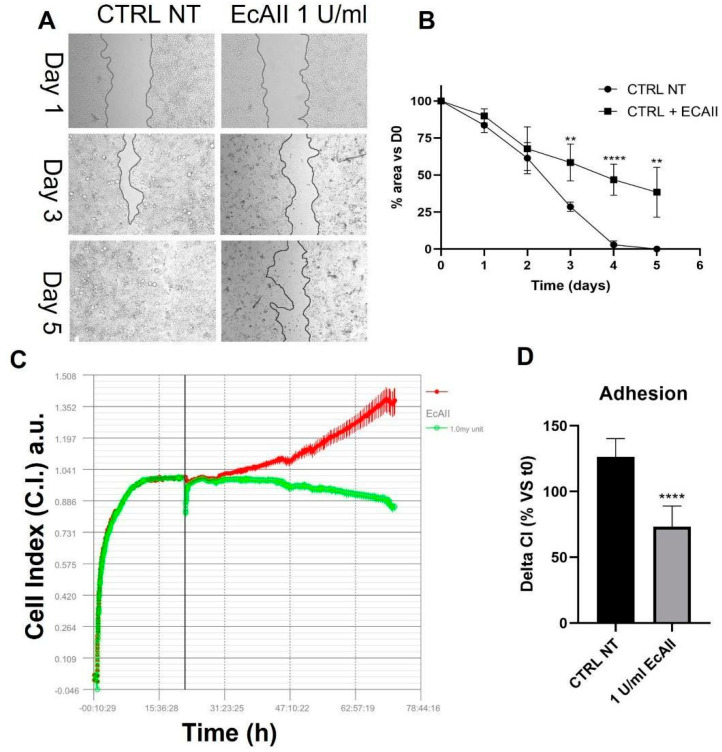
Proliferation/migration (wound healing assay) and adhesion (iCELLigence real-time cell analysis, RTCA) for Caco-2 cells treated with 1 U/mL EcAII. (**A**) Representative images of wound healing assay after 1, 3 and 5 days from Ibidi insert removal. Untreated control (CTRL NT) on the left and sample treated with EcAII 1U/mL on the right. (**B**) Percentage of the septum area not covered by cells as a function of time (from day 1, 100%, to day 5) after insert removal in the wound healing assay, for control and EcAII-treated Caco-2 cells. (**C**) Representative sensorgram of Caco-2 cells w/o EcAII (red line and dots) and with EcAII (green line and dots). Each data line represents the average of eight wells. The vertical line corresponds to the time of addition of EcAII (24 h). (**D**) Quantification of delta cell index (dCI%) obtained from RTCA 72 h after seeding without (black bar) or with (gray bar) 1 U/mL EcAII. dCI is expressed as the CI % variation at the endpoint with regard to the beginning of treatment (24 h after seeding). Data reported are mean ± SD, obtained from at least three independent experiments. Statistical significance (multiple Student’s *t*-test) was calculated comparing the w/o and w/EcAII conditions at all other equal conditions, and is reported as follows: ** *p* < 0.01, **** *p* < 0.00001.

## Data Availability

Not applicable here.

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
