# Peer review of "Asparagine and Glutamine Deprivation Alters Ionizing Radiation Response, Migration and Adhesion of a p53null Colorectal Cancer Cell Line"

_ijms, 2023, doi:10.3390/ijms24032983_

Round 1
Reviewer 1 Report
The manuscript by Guardamagna et al investigates the effect of the combination of L-Asparaginase and radiation on the colorectal cancer cell line Caco-2. The study indicates a combinatorial effect in decreased cell survival when L-ASNase treatment is carried out prior and directly after radiation though the effect seems to be additive rather than synergistic and the treatment alone appears to have a considerable effect on clonogenic survival. The study then aims to decipher the mechanism of the combined toxicity using biomarkers for DNA damage and repair, autophagy, cell migration and adhesion and cell cycle regulation. I believe this study to be well planned and a steppingstone for further studies on the combination of these two treatments including additional cell lines of different genetic background and toxicity in normal cells. I only have two comments
1. While the use of Chk-2 and cyclin B status is useful, the authors should use flow cytometry analyses of the cell cycle to obtain more quantitative data on the effect of the combination relative to each treatment for conclusive data.
2. The authors should use the number of foci per nuclei for the correct measurement of the y-H2AX levels over time which is the most sensitive method to measure y-H2AX. The use of intensity measurements using image J is an imprecise method which could be affected by the quality of the images. For the higher doses it is possible to use the total area and divided it by the average size of foci to get an estimate of foci per cell. Measurement of intensity from western blot images are at best semi quantitative and it is not clear why they are used for these time points. Further this could be confounded by the intense pan y-H2AX in the apoptotic cells at the late time points. To use y-H2AX mean fluoresce intensity, flow cytometry should be used for true measurement of fluorescent intensity per cells. As it is in the manuscript now, no real dose-dependent increase due to IR is detected at lower doses and early time points, and the non-treated values at different time points are very different pointing to the lack of measurement sensitivity.
Reviewer 2 Report
Figure 1, is the synergy seen with radiation and L-asparaginase treatment specific to p53 null status? Can the authors test this hypothesis by p53 reexpression in CaCo cells to see if it rescues this phenotype? I feel this is an important question to evaluate if stratification of patients based on p53 status is necessary.
Figure 2D, 5 Gy the quantification increase in p-Chk2 signal is not convincing if you look at the blot presented at Figure 2A. Can the authors include cell cycle analysis by FACS to demonstrate G2/M checkpoint activation? What happens to G1/S transition? In this regard, FACS analysis would give a complete picture.
Figure 3E labeling, both are labelled w/EcAII. Again, the blots here are not convincing. From the quantification of gH2AX signal in the western blot, there is no difference in gH2AX signal between wo /EcAII and w/EcAII. Can the authors validate it with immunofluorescence? Does this data mean the cells repair DNA damage at the same rate wo /EcAII and w/EcAII treatment?
Reviewer 3 Report
In the present article by Guardamagna et al., the authors describe their data investigating the response of p53 deficient colon carcinoma cells (Caco-2) to L-Asparaginase (L-ASNase/ EcAII) comprising asparagine and glutamine metabolism and its potential radiosensitizing effect. The authors investigated various biological endpoints such as clonogenic survival, G2/M arrest, DNA damage, and migration. The authors' experimental results show that EcAII alone may already have antitumorigenic effects on p53-deficient colon carcinomas with further potential for radiosensitization. Studies investigating multimodal therapeutic strategies with novel molecular agents are important to improve clinical outcomes in refractory tumor entities.
However, some major comments should be considered before publication.
1) At first, a major drawback is that the work was conducted with a single cell line only. The results should be confirmed in at least one additional colon carcinoma cell line.
2) Lines 52-54: The desired effect is synergy beyond additivity. These terms should be mentioned and introduced here. They should be familiar to the reader of this journal and are also used later in the manuscript (discussion).
3) Line 78: In addition to norm-fractionation, the authors should also mention hypofractionation (short course with higher single doses).
4) Figure 1: Why do the authors not show classic survival curves in the semi-log plot? This would also help in the assessment and differentiation between additive or synergistic effects. This is a bit neglected in the manuscript anyway.
A reduction of the plating efficiency (0Gy) could be shown separately.
Does a reduction in plating efficiency t correlate with the adhesion results in Figures 6 C and D? This should be considered and discussed.
5) Line 112: In the figure legends as well as the Material and Methods section the authors mention they have performed three independent experiments. Have these experiments been performed with biological, or technical replicates? Please provide this information for all assays.
6) Line 118 and the following concerning the G2/M assay: why did the authors not apply conventional cell cycle measurements by flow cytometry?
7) Line 124 and the following: Too much methodological information is found in the results section, often redundantly regarding the corresponding figure legends and Material and Methods sections. This should be reduced.
Altogether, the data on pChk2 and cyclinB1 is questionable. For a p53-negative tumor cell line with a high proliferation index, one would expect a G2/M arrest already 6-8h after radiation exposure. In the 5Gy w/EcAII setting, there is a lower level of Cyclin B1 after 5Gy compared to the 5Gy w/o. Does this indicate an abrogation of the G2/M arrest? What do the authors want to show? A prolonged G2/M arrest by concomitant EcAII treatment or abrogation of the G2/M arrest promoting mitotic catastrophes and replicative cell death? Overall, there are generally only minor changes that seem to be nonsignificant for any changes up to 48h for doses below 5 Gy.
Regarding densitometric protein quantification, I would expect to show a loading control (e.g, actin, vinculin, or GAPDH,…) for analysis of phosphorylated isoforms, too. This would be much more convincing and provide some general information.
8) Line 164 gH2AX formation: Why did the authors apply the pre-treatment protocol (72h w/EcAII) for gH2AX signals and did not stick to the post-treatment protocol? Why did the authors not quantify gH2AX foci which is a much more sensitive method? Quantification of mean immunofluorescence per cell, respective nucleus, is affected by many factors. The methodology should be described in more detail in the Material and Methods section, such as background corrections, etc.
Later time points concerning residual DNA damage at 6h, 24h, and 48h post-IR were analyzed by gH2AX quantification by Western blotting and with the post-treatment regime for EcAII. Here again, as for pChk2 and CyclinB1 western blots, there are only very minor changes and the data is not very convincing. Quantification of gH2AX foci for all time points of analysis should have been performed using the same post-treatment protocol.
9) Line 454, Western Blot: Detailed information on densitometric quantification is lacking.
Round 2
Reviewer 1 Report
Comments on the revised manuscript by Guardamagna et. Al.
Regarding comment 1, the authors amendment to the data and manuscript is satisfactory.
Regarding comment 2 and 3 on the measurement of pan-nuclear gamma-H2AX levels as a marker for ROS-induced ssDNA breaks, the response is not acceptable for reasons outlined below:
1. The paper on phosphorylation of H2AX histones resulting from cited by the authors (PMC1502549) is a unique case for UV induced damage in fibroblasts and to my knowledge the pan-nuclear staining of gamma-H2AX has not been generalized as a biomarker for ssDNA damage in the literature. The level of ssDNA breaks is routinely measured using the comet assay.
2. Numerous papers attribute pan-nuclear gamma-H2AX staining to lethal replicative stress and apoptotic cells also reiterating that the methods used here cannot be used as a marker for ssDNA break and moreover these cells need to be excluded while measuring DSBs, which is not possible with western blots. On the point of the use of this less appropriate method (western blot), the authors response of ’a strategy to integrate data obtained with different techniques’ would only be acceptable if this strategy was used for all conditions and time points and not in a selective manner (i.e. all methods for all data points).
While I don’t believe that the interpretation and design for this section impacts the overall merit of the article, it is incorrect to present this data as a measurement of ssDNA damage and needs to be amended as the authors see fit.
Reviewer 3 Report
All my concerns and comments have been taken into account and extensively edited by the authors. I congratulate the authors on this thorough work and now recommend the publication of the article in its present form.
Author Response
We thank the reviewer for his/her contribution to improve on our manuscript.
Round 3
Reviewer 1 Report
Although I maintain that measurement of the gamma-H2AX signal in the way carried out in this paper is not appropriate for the 'measurement' of ssDNA breaks and the publications in support of this that the authors provided also use additional data to assign this type of activation to ssDNA breaks, and do not measure the MFI, rather number of cells, given that the manuscript is now written in a way to present this as indirect evidence of ssDNA breaks and that this is not critical for the impact of the article, I believe that it can be accepted in its present form.